# The impact of agricultural industrial agglomeration on farmers' income: An influence mechanism test based on a spatial panel model

Yi Ding [ORCID] *

School of Management, Sichuan Agricultural University, Chengdu, China

* oneho@qq.com

## Abstract

Recently, China has exerted great efforts to develop agro-industrial agglomerations and optimize the agricultural industry's regional distribution to increase farmers' income. This study posits that agro-industrial agglomeration should be "dual body," provides a theoretical framework for agro-industrial agglomeration's effects on farmer income, and uses a spatial panel model to prove its influence on farmers' income and the spatial spillover effect. The results show that agro-industrial agglomeration in a specific region significantly impacts farmers' income and also has a spillover effect on income in adjacent regions. Further research shows that, contrary to traditional agglomeration theory, agricultural industry agglomeration has little impact on farmers' agricultural production but primarily promotes the establishment and development of agricultural organizations, thus improving farmers' income. Finally, the paper discusses the positive and negative effects of agro-industrial agglomeration on the Global Sustainable Development Goals(SDGs), and proposes some useful suggestions.

## 1. Introduction

In September 2015, the United Nations adopted "Changing our World: The 2030 Agenda for Sustainable Development," marking the official establishment of the Global Sustainable Development Goals (SDGs). The SDGs aim to solve the development problems of social, economic, and environmental dimensions in an integrated manner, via a sustainable development path. Subsequently, the traditional concept of development has experienced subversive changes, and current consensus is that human development should be sustainable. As an important approach to industrial development, agro-industrial agglomeration is conducive to increasing farmers' income, and is of great significance to the SDGs, such as alleviating poverty (SDG1), preventing hunger (SDG2), promoting sustainable economic growth and full employment (SDG8), and constructing infrastructure to promote innovation (SDG9). Recently, China has been striving to develop an agricultural industry cluster, optimize the agricultural industry's regional layout, and drive farmers to increase their income. From 2018 to 2022, for five

**Data Availability Statement:** The data underlying the results presented in the study are available from: (1) The CSMAR database: https://www.gtarsc.com/ (2) The CNKI database: https://www.

cnki.net/ (3) National Bureau of Statistics: http://www.stats.gov.cn/tjsj./ndsj/.

**Funding:** The author(s) received no specific funding for this work.

**Competing interests:** The authors have declared that no competing interests exist.

consecutive years, the Central First Document has mentioned the need to develop special agricultural products based on resource advantages, create an agro-industrial chain, and establish a sound mechanism for farmers to share the chain's value-added income, making it clear that agro-industrial agglomeration is an important way for farmers to increase their income.

Since this phenomenon was first proposed by Marshall in 1890, many scholars have argued that industrial agglomeration plays an important theoretical and practical role in reducing production and transaction costs, promoting technological innovation, enhancing information exchange, and improving labor productivity and industrial chains [1–15]. Scholars believe that agro-industrial agglomeration is subordinate to general industrial agglomeration and apply it in related theories. Many scholars have proven that agro-industrial agglomeration plays an important role in enhancing economies of scale, promoting technological innovation, reducing transaction costs, improving industrial chains, and increasing production efficiency in agricultural production [16–21]. However, these theories are primarily applicable to agribusinesses and not to individual farmers. For example, farmers are weak and mostly risk-averse. They generally do not innovate production technology, and the innovation effect brought by industrial agglomeration does not apply to them, as most only engage in production, and intermediaries or downstream enterprises buy agricultural products at their doorsteps. Overall transaction costs are low compared with those of enterprises; as a result, industrial agglomeration does not reduce farmers' transaction costs. Reflecting the national condition of "big country and small farmers," China has more than 90% small farmers [22], and even if there is a complete industrial chain, most farmers seldom participate in other links, such as processing and marketing.

The literature has overlooked an important difference between industrial agglomeration and agricultural agglomeration, namely, industrial agglomeration has a single business subject, while agricultural agglomeration has two business subjects: enterprises and farmers. Since industrial employees exist within enterprises, they are not independent business subjects. Studies on industrial agglomeration focus on the enterprise level, and the findings are primarily applicable to agglomerated enterprises. In agro-industrial agglomeration, farmers have scarce land and labor resources. They produce primary agricultural products and engage in mass production by establishing family farms. Therefore, farmers, as independent business subjects with auto-sustainability, merit the same status as enterprises and should be treated as subjects of agro-industrial agglomeration. In many rural areas, where mechanization and modernization remain undeveloped, farmers' status as the primary production force in such agglomeration cannot be ignored. Accordingly, this study believes that there should be two primary bodies in agricultural industry agglomeration: farmers and enterprises.

Because of the poor understanding of the dual subjects of agro-industrial agglomeration, the literature pays less attention to farmers in agglomeration; therefore, agglomeration theory applies to agglomeration enterprises, but not to farmers. The fundamental reason is that farmers' and enterprises' status are equal in agro-industrial agglomeration, but their resource endowments differ substantially. Enterprises have strong capital, talent and technology and rapidly benefit from various industrial agglomeration effects. For historical reasons farmers in China, by contrast, have small and fragmented arable lands resulting in minor and scattered production and operations. Their returns from agricultural production are generally lower. This has caused urban migration among capable youth, leaving agricultural production to aged and part-time labor. Most laborers remaining in rural areas are women, children, and elderly with low education levels. Their general ability to adopt new technologies and varieties is also low. With poor resource endowments in capital and labor, farmers find it difficult directly enjoy industrial agglomeration effects such as economies of scale or technological innovation.

Except for Wang YR and Liu [23] and Yang LJ [24], who argue that agro-industrial agglomeration has no significant effect, other scholars believe that it improves farmers' income. This may be due to the small amount of data and limited representativeness of the two scholars' models as well as the pre-2010 data, when the degree of agro-industrial agglomeration was low. Industrial agglomeration has a threshold effect on farmers' income and has less impact when the degree of agglomeration is low [25]. Both used impulse response analysis, which cannot exclude other factors from interfering with the impact of farmers' income, so the research results were insufficiently precise. In another study, Tang LJ and Zeng [26] used panel data of nine citrus agglomerations in Jiangxi Province to build a regression model to prove that such industry agglomeration can attract the convergence of land, labor, capital and technology to increase local economic benefits and drive the increase in area farmers' income. Using cross-sectional aquaculture data from Jiangsu, Zhejiang, and Guangxi provinces in 2016, Li BW et al. [25] constructed a regression model to prove that agglomeration significantly contributes to increasing farmers' income when the agglomeration degree surpasses the threshold value of 0.7619. Using cross-sectional data from the dominant vegetable production areas in the Yellow and Huaihai Seas and the Bohai Rim in 2016 and using geographical distance as the spatial weight matrix, Zhang ZX and Mu [20] argued that the industrial agglomeration effect significantly increases local farmers' and neighboring farmers' incomes, and agglomeration's spillover effect on farmers' income has the characteristic of decreasing geographical distance.

A literature review reveals that: (1) most recent studies used cross-sectional data, while industrial agglomeration is a slow process, and its effect also occurs slowly; therefore, the use of panel data can better reflect gradual accumulation and increases in its impact on farmers' income.(2) The literature has ignored the spatiotemporal effects of agro-industrial agglomeration on farmers' income; Zhang ZX and Mu [20] examined agglomeration's effects on farmers' income using a spatial model but was limited by cross-sectional data and ignored the temporal effects. He used geographic distances as spatial weights and ignored the effects of economic distance. In this study, the spatial panel model was used to consider agglomeration's spatiotemporal effects on farmers' income, while the economic-geographic nested matrix served as a spatial weight matrix to more comprehensively examine agglomeration's spatial effects. Moreover, this study provides an analysis of heterogeneity. (3) Studies agree that agglomeration effects, such as economies of scale and technological innovation, directly and significantly affect farmers' income. However, we posit the concept of the "double main body" of agro-industrial agglomeration, arguing that the main body of farmers cannot directly benefit from industrial agglomeration effects because of poor resource endowments. The increase in farmers' income is primarily influenced by other subjects such as enterprises and cooperatives; that is, income increases through the indirect influence of industrial agglomeration, and this view is supported by empirical research.

The contributions of this paper are as follows: (1) The difference between industrial agglomeration and agricultural agglomeration is pointed out, and the concept of "double subject" of agricultural agglomeration is proposed; (2)This study clarifies the main ways in which agricultural industrial agglomeration acts on individual farmers, including three types: promotion, traction and policy, and proposing that traction should be the most important way; (3) The spatial panel model is used to prove the three ways, and traction is confirmed as the most important way. This study not only helps reveal the interest linkage mechanism between agro-industrial agglomeration and farmers, but also helps to provide a theoretical reference and practical basis for the public sector to formulate policies promoting farmers sharing in the dividends of industrial development to achieve the ambitious goals of regional agricultural economic growth and rural revitalization.

The remainder of this paper is organized as follows: Section 2 constructs a theoretical framework of the impact of agricultural industry agglomeration on farmers' income and proposes research hypotheses. Section 3 introduces the research proposal. Section 4 examines the spatiotemporal evolution of agro-industrial agglomeration in China, while Section 5 analyzes its related effects on farmers' income. Section 6 verifies the primary paths of agro-industrial agglomeration's effects on farmers' income. Section 7 concludes with policy recommendations.

## 2. Theory and hypotheses

### 2.1. Agro-industrial agglomeration promotes farmers' income

Since farmers in China have important production factors of land and labor, have long engaged in agricultural production as family units, and can establish large-scale production, family farms can be regarded as self-sustaining and independent operating entities. The main subject of agricultural labor in China remains farmers, and their important position in industrial agglomerations is self-evident. The main bodies of agro-industrial agglomeration should be the farmers and enterprises in an area. Agglomeration's impact on farmers' incomes should be divided into direct and indirect. The direct impact is agro-industrial agglomeration's effect on farmers' agricultural production, prompting an increase in agricultural business income. The indirect impact is the influence of agricultural enterprises, cooperatives, government, and other regional organizations, which results in farmers obtaining additional non-farm income. In summary, the role of agro-industrial agglomeration in farmers' income primarily in the areas of promotion, traction, and policy. The theoretical derivation is shown in Fig 1.

Agro-industrial agglomeration's role in promoting farmers' income refers to its role in promoting agricultural business income, which is the primary income source, specifically:①agricultural industry agglomeration promotes technological innovation and information diffusion in the agglomeration area, so that farmers can more easily accept and adopt new technologies and new varieties, thus increasing related income. ② Agro-industrial agglomeration concentrates agricultural production and sales facilitating creation of landmark brands with regional characteristics, increasing product competitiveness; higher brand awareness leads to increased

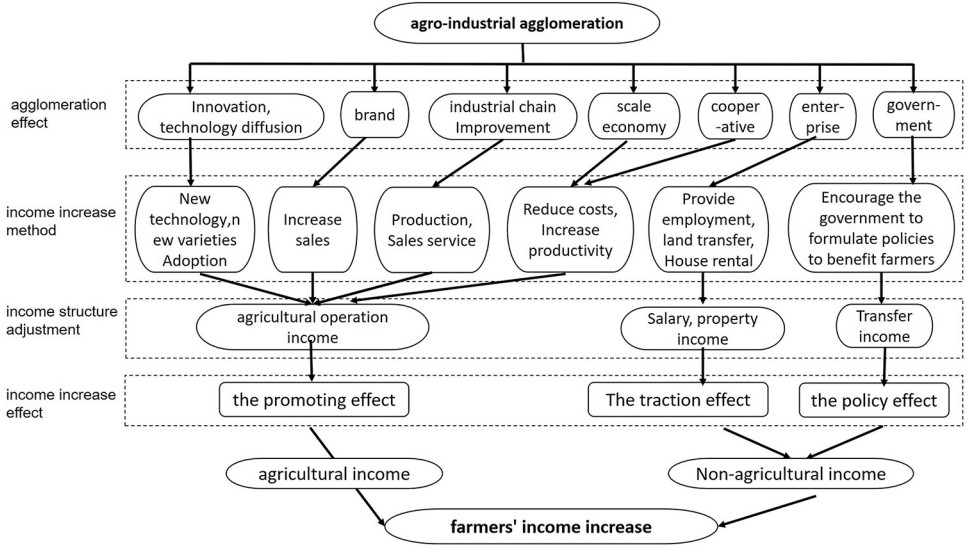

**Fig 1. The effect mechanism of agro-industry agglomeration on farmer'income.**

demand, enabling farmers to expand production scale and producing scale effects, thus reducing production and transaction costs and improving efficiency. ③Agro-industrial agglomeration can improve the industrial chain so that farmers have access to professional production services during the busy season. In the marketing stage, farmers can make use of external sales-related services such as rough processing, frozen storage, packaging, and cold chain transportation to expand production scale, extend sales time, and increase sales. In addition, some part-time farmers can manage agricultural production while working. ④ Agro-industrial agglomeration promotes the establishment of professional farmers' cooperatives, which can save costs in materials procurement and improve production efficiency through group technical training. Simultaneously, cooperatives can transfer land, employ farmers in production, and provide services such as rental of agricultural machinery. Therefore, the establishment of professional farmer cooperatives can simultaneously increase farmers' incomes from agricultural operations, wages, and property.

The traction effect of agro-industrial agglomeration on farmers' income refers to expanding farmers' income sources and create additional non-farm income sources. Agro-industrial agglomeration's traction effect on farmer income primarily promotes the creation and development of agro-enterprises in the agglomeration area, thus increasing farmers' wage and property income; specifically, farmers may earn wage income as firm employees. As agro-enterprises expand production scale, farmers may provide services such as land transfer, housing leases, and large machinery and equipment storage, thus creating property income.

Agro-industrial agglomeration's policy effect on farmers' income refers to causing the development of the local agricultural economy, stimulating regional economic growth, and increasing tax revenue and employment, thus drawing the attention of management departments. This promotes formulating preferential policies for the agglomeration area and investing additional resources in infrastructure construction. Therefore, industrial agglomeration can increase farmers' transfer of income.

Agro-industrial agglomeration also has a spillover effect on farmers' income. In terms of direct impact, it concentrates agricultural products in local production and sales, which also drives the production and sales of similar agricultural products in the surrounding area, promoting local innovative production technology, which can be more rapidly acquired by farmers in the surrounding area. Agglomeration improves the local agro-industrial chain, and farmers in the surrounding area also benefit from certain production and sales services during the busy farming season. In terms of indirect impact, agglomeration enhances the strength of local agricultural enterprises and promotes large-scale production. The enterprises also absorb agricultural labor from the neighboring areas, which drives the growth of neighboring farmers' wage income. According to the law of spatial correlation proposed by Waldo Tobler, the closer the distance, the greater the correlation between features. With the continuous improvement of the market economy, communication conditions, and transportation capacity, the mobility of production factors, such as labor, capital, and technology, increases, and the spatial spillover effect of agglomerations also affects the income growth of farmers in neighboring regions.

Accordingly, this study proposes hypothesis 1: agro-industrial agglomeration can promote farmers' income growth and have spatial spillover effects.

## 2.2. The influencing mechanism

The essence of industrial agglomeration is the relative concentration of interrelated business organizations in a specific field in a geographic space. Because of the concentration of geographic space, rapid and high-quality information exchange exists between related businesses, thus reducing transaction costs, promoting technological innovation, and forming economies

of scale. Agglomeration primarily promotes business organizations to form large-scale mass production and thus enhance market competitiveness. Therefore, agglomeration mainly affects organizations with enhanced strength and scale; that is, the agglomeration of the agricultural industry primarily promotes the development of agricultural enterprises and other large-scale organizations, while farmers hardly benefit from industrial agglomeration directly because of their weaknesses.

Therefore, the mechanisms analyzed above that increase farmer income differ. In the case of "big country and small farmers" and farmers' financial weakness, the traction effect should be the primary influence mechanism. Agro-industrial agglomeration promotes the continuous development of agricultural enterprises, farmers' cooperatives, and other agricultural business organizations in a region, enhancing their income. The advantages of agro-industrial agglomeration directly affect regional enterprises, which increases their number in the region and intensifies competition among them. Based on fierce competition and their own development requirements, enterprises are bound to cooperate with area farmers because of their land and labor assets, and thus enterprises can drive farmers' production and lead them to improve their income. The traction effect mechanism is shown in Fig 2.

**2.2.1. Agro-industrial agglomeration establishes and accelerates the development of various agricultural business organizations.** According to industrial agglomeration theory, its most direct and strongest impact is on agricultural enterprises. It concentrates a considerable number of related enterprises in a specific area, which means that a large amount of skilled labor, raw materials, and other production factors gather in a specific area. This simultaneously reduces the transportation costs between enterprises and entry barriers for new enterprises to enter the industry. Simultaneously, the gathering of core enterprises attracts supporting enterprises and service institutions, further reducing the difficulty of starting a business. For example, agglomeration attracts the entry of financial institutions, making it easier for new enterprises to obtain financing. In addition, the geographical proximity of industrial clusters makes firms better rooted in local culture and enables entrepreneurs to form unified values, which positively influences firms' social capital, enhances trust and cooperation among firms, and makes entrepreneurship more likely to succeed. Thus, Fujita M and Thisse [27] argue that industrial agglomerations be considered incubators and stabilizers for firm generation. Mashell A [28] also argues that industrial agglomerations attract economic activities and achieve efficient resource allocation. Numerous scholars have argued that industrial agglomeration helps establish new enterprises [29–34].

It not only attracts new enterprises but also promotes their accelerated development. Owing to geographical proximity, the communication cost between enterprises decreases, which promotes information exchange and generates a knowledge spillover effect. This not

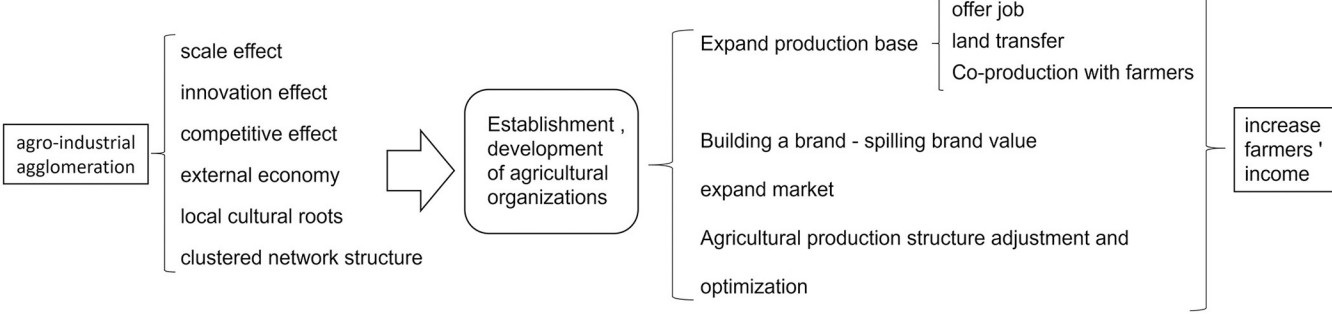

**Fig 2. Agro-industrial agglomeration affects farmer income through agricultural organization.**

only facilitates enterprise innovation, but also promotes division of labor and collaboration, accelerating enterprise development. Industrial agglomeration brings a complete industrial chain, making it easier for enterprises to make decisions on independent production or outsourcing, achieving the optimal combination of transaction and management costs and reducing operating costs. As the number of enterprises in the agglomeration area increases, it intensifies competition among them, and the resulting competitive effect forces accelerated development. owing to the dependence on the region, the industrial resources released by failed enterprises will not spread outside the agglomeration area and can still be used by area enterprises [35].

Agro-industrial agglomeration not only acts on agricultural enterprises but also contributes to the creation and development of local farmers' cooperatives and agricultural associations. When the number of farmers producing similar agricultural products in the agglomeration area increases, it leads to the formation of organizations such as farmers' cooperatives and agricultural associations in the area. With the increasing number of enrolled farmers, cooperative organizations slowly strengthen, and as bargaining power in the purchase of raw materials such as pesticides and seeds increases, production costs decrease. In the production process, larger cooperatives develop capabilities to regularly hire agricultural experts to provide technical training to members, to purchase large agricultural machinery, and even create their own brands to obtain brand premiums. During product sales, cooperatives are more likely to unify management and reach agreements to maintain stable product prices because they unite a large number of producers, and they can sign cooperation agreements with agricultural enterprises to expand sales.

**2.2.2. Agricultural business organizations are bound to drive farmers' income growth based on their own competition and development.**   First, to develop, enterprises must expand the scale of their production bases and stabilize the sources of agricultural products. In reality, many agricultural enterprises, through "company + farmer","company + cooperative + farmer","company + base +farmer"and other forms of cooperation, will enhance farmers wage and agricultural business income. Agricultural enterprises provide farmers with seeds and training in planting techniques based on the need for standardized production and stable product quality. Farmers acquire advanced production techniques through learning and observation. Agribusinesses also have the strength to establish proprietary research and development (R&D) departments or cooperate with research institutes to continuously innovate production technologies and develop new varieties, which are then promoted and used by farmers for cooperative needs, driving them to continuously improve their production technologies. In another form, agricultural enterprises transfer farmers' land and then employ them as production workers so that they can obtain property and wage income. Owing to land scarcity within the agglomeration area, the price of land transfer within the area is usually higher than that outside the area, which enables farmers to obtain higher returns. Either way, agribusinesses cooperate with farmers, increasing their incomes.

Second, as most farmers lack commodity awareness and market management ability, it is very easy to blindly follow agricultural production trends. This makes agricultural products homogenized and identical, resulting in an imbalance between supply and demand and damaging agricultural income. Agricultural enterprises, by contrast, have strong market perception and risk management capabilities and can make timely production adjustments according to market demand to ensure the basic stability of prices; thus, farmers in the agglomeration area can enhance income while increasing production. In addition, farmers are almost powerless regarding expanding market demand. However, enterprises can expand demand by constantly innovating agricultural products and opening sales channels. For example, the town of Anyue, Sichuan, is one of the largest lemon industry clusters in China. Agribusinesses in the cluster

convert lemons into masks, vitamin C lotions, tablets, etc., which greatly expands market demand. Another example is the Wanyuan tea agglomeration in Dazhou, Sichuan, where the enterprise Guozhu Agricultural Development Co., Ltd. converts tea into matcha, green tea, green tea powder, and other products, and exports Wanyuan tea. Enterprises in the agglomeration area based on their own development needs, efforts to expand market demand and sales channels, which will also drive the production and sales of farmers in the agglomeration area.

Owing to the high degree of homogenization of primary agricultural products, product competition is concentrated in brand competition. It is difficult for farmers to brand their agricultural products because of the limitations of culture and economic strength. Agricultural enterprises, by contrast, can build and market their brands based on their development needs. When there are more high-profile brands in the agglomeration area, it also produces a brand spillover effect and enhances the visibility of local landmark products. Farmers directly benefit from the enhancement of the visibility of landmark brands and will also receive brand premiums to enhance unit prices and sales, thereby increasing income.

According to the principle of corporate nature, agricultural enterprises have a strong internal drive to increase farmers' incomes because of their developmental needs. Corporate enterprises are characterized by the "union of capital," and shareholders can only obtain residual income after deducting company costs and expenses, and they must bear all the risks of firm operation. With the intensification of competition among enterprises in the agglomeration area, this system involves a developmental impetus. Agribusinesses are responsible for the creation and operation of agricultural industry, supply, and value chains, which inevitably involve core aspects such as base construction, product quality upgrading, technological innovation, marketing capacity enhancement, and social resource development. Agricultural enterprises can enhance their core competitiveness and achieve sustainable development only by creating and operating these core links. Based on the strong internal driving force of their development, agricultural enterprises vigorously drive farmers to simultaneously increase their income.

In addition to enterprise agglomeration driving farmers' income growth, regional farmers' professional cooperatives and production associations can also play a positive role in driving farmers. The main purpose of these cooperatives and associations is to unite farmers, reduce production costs, promote technology dissemination, increase bargaining power in the market, and improve competitiveness so that they can enhance income.

Accordingly, this study proposes hypothesis 2: The impact of agro-industrial agglomeration on farmers' income is mainly traction effect. Agro-industrial agglomeration leads to the establishment of various agricultural organizations led by enterprises and accelerates their development of these organizations. In turn, various agricultural organizations drive farmers' production and income based on their development needs, that is, various agricultural organizations led by agricultural enterprises play an intermediary role in agro-industrial agglomeration's impact on farmers' income.

## 3. Methods

### 3.1. Data sources

Limited by data representativeness and availability, this study selected 2013–2019 panel data for 30 provinces, autonomous regions, and municipalities in China, except for Hong Kong, Macau, Taiwan, and Tibet. The data were obtained from the CSMAR, EPS, China Population and Employment Statistical Yearbook, China Statistical Yearbook, China Rural Statistical Yearbook, China Financial Yearbook, National Bureau of Statistics, statistical yearbooks of each province, and relevant government websites. The data set used in this paper can be seen in S1 Appendix.

### 3.2. Variable selection

**3.2.1. Dependent variable.** This study used farmers' income as the explained variable, which is measured by rural residents' per capita disposable income in the EPS database. The per capita disposable income of rural residents reflects farmers' income after primary distribution and redistribution, which reflects the overall situation of farmers' actual income. As a result of the National Bureau of Statistics' (NBS) reform of the integrated urban-rural household income and expenditure survey, the previous "net income of rural residents" has been replaced by "per capita disposable income of rural residents" since 2013. To ensure the continuity and comparability of data, this study used data from 2013 and later.

**3.2.2. Independent variable.** The explanatory variable here is the concentration of agricultural industries in each province. This study adopts the locational entropy method, which is more commonly used in the literature, to measure the degree of agro-industrial agglomeration. Location entropy was first proposed by P. Haggett and applied in location analysis, which is a response to the degree of a certain industry's concentration in the area where it is located and is particularly suitable for analyzing the degree of industrial agglomeration from a regional perspective. The specific calculation is as follows:

$$LQ_i = \frac{m_i/m}{M_i/M}$$

Where $m_i$ denotes the output value of the provincial agricultural industry, $m$ denotes the total output value of all provincial industries, $M_i$ denotes the output value of the country's agricultural industry, and $M$ denotes the total output value of all country industries. In this study, LQ was used to indicate the degree of agglomeration of agricultural industries in each province. LQ>1 indicates a higher degree of agglomeration of agricultural industries and LQ<1 indicates a lower degree of agglomeration of agricultural industries in the province.

**3.2.3. Control variables.** Five control variables were selected: the level of local economic development, rural fixed asset investment, agricultural loans, level of agricultural modernization, and rural human capital. Among them, the level of agricultural modernization is measured by the total power of agricultural machinery per capita in rural areas; the formula of rural human capital is: rural human capital = (the number of population with primary school education in the sample*6 + the number of population with middle school education in the sample*9 + the number of population with high school education in the sample*12 + the number of population with specialist and above education in the sample*16)/the number of sample population above 6 years old, to reflect rural residents' education level.

**3.2.4. Mediating variables.** According to the theoretical derivation, the mediating variable selected here is the rural business organization. Since the number of agricultural enterprises in each province is unavailable, and there are a large number of "shell societies" of farmers' cooperatives in each province, it is not suitable to use the number of agricultural enterprises and farmers' cooperatives directly. In this study, we used the sum of the number of national-level leading enterprises and national-level farmers' cooperatives to measure the number of rural business organizations. National-level leading enterprises and farmer cooperatives are selected by the National Ministry of Agriculture and Rural Affairs on a regular basis, and the selected list is dynamically adjusted according to uniform criteria. The data were collected manually based on a list published on the official website of the Ministry of Agriculture and Rural Affairs.

To reduce the effect of heteroskedasticity, the relevant model variables were taken as logarithms; to avoid the interference of extreme values, the data were subjected to a tailing process from 1% to 99%. The variables and their descriptive statistics are shown in the Table 1.

**Table 1. Variable table.**

| Category | Variable | Symbol | Definition | Mean | SD | Min | Max |
|---|---|---|---|---|---|---|---|
| Dependent variablble | farmers' income | Income | Per capita disposable income of rural residents | 9.424 | 0.343 | 8.629 | 10.41 |
| Independent variable | agglomeration degree of agricultural industry | Gather | It is calculated by the location entropy method | 1.198 | 0.639 | 0.038 | 3.287 |
| Mediating variable | number of agricultural business organization | Firm | The sum of the number of national leading enterprises and national farmer cooperatives | 3.531 | 0.436 | 2.639 | 4.454 |
| Control variables | the level of local economic development | GDP | Per capita GDP of each province | 10.88 | 0.41 | 10.05 | 12.01 |
| | rural fixed asset investment | Invest | Per capita rural residents investment in fixed assets | 7.363 | 0.556 | 4.7 | 8.485 |
| | agricultural loans | Credit | Loans to rural households of each province/rural population of each province | 9.204 | 0.677 | 6.62 | 11.4 |
| | level of agricultural modernization | Machine | Total power of agricultural machinery in rural areas of each province | 7.684 | 1.123 | 4.543 | 9.499 |
| | rural human capital | Edu | (the number of population with primary school education in the sample * 6 + the number of population with middle school education in the sample * 9 + the number of population with high school education in the sample * 12 + the number of population with specialist and above education in the sample * 16)/the number of sample population above 6 years old | 7.818 | 0.601 | 5.878 | 9.801 |

## 3.3. Model framework

**3.3.1. Spatial effect model.** In order to study the impact of agro-industrial agglomeration on farmers' income, the basic model is first established (Eq 1).

$$income_{it} = \alpha + \beta_1 gather_{it} + \beta_2 C_{it} + \varepsilon_{it} \tag{1}$$

Where $income_{it}$ denotes the per capita disposable income of rural residents; $gather_{it}$ denotes the degree of agro-industrial agglomeration; $C_{it}$ is the control variable, which is the other factors affecting farmers' income; $\alpha$ is the constant term; $\beta_1$ and $\beta_2$ are the regression coefficients; and $\varepsilon_{it}$ is the random error term.

Agro-industrial agglomeration not only has a intra-regional impact, but also may have a cross-regional impact. Spatial lag regression model (SLR), Spatial error model (SEM) and Spatial Durbin model (SDM) are established to explore the spatial effects of agro-industrial agglomeration on farmers' income.

Spatial lag regression model (SLR):

$$income_{it} = \rho W_{ij} income_{jt} + \beta_1 gather_{it} + \beta_2 C_{it} + \varepsilon_{it} \tag{2}$$

Spatial error model (SEM):

$$income_{it} = \beta_1 gather_{it} + \beta_2 C_{it} + \mu_{it} \tag{3}$$

Spatial Durbin model (SDM):

$$income_{it} = \rho W_{ij} income_{jt} + \beta_1 gather_{it} + \beta_2 C_{it} + \beta_3 W_{ij} gather_{jt} + \beta_4 W_{ij} C_{jt} + \varepsilon_{it} \tag{4}$$

In Eqs (2) ~ (4): $\mu_{it} = \lambda W_{ij} \lambda_{jt} + \varepsilon, \varepsilon \sim N(0, \sigma^2 I)$, $W_{ij}$ is the spatial weight matrix, $\lambda$ is the spatial error coefficient, $\sigma$ is the standard deviation of normal distribution, and I is the identity matrix. $W_{ij} income_{jt}$ is the spatial lag term of the explained variable; $\rho$ is the correlation coefficient, reflecting the spatial linkage degree of the observed values of the explained variables in different regions. $W_{ij} gather_{jt}$ was the spatial lag term of the explanatory variable. $W_{ij} C_{jt}$ is the spatial lag term of the control variable; $\beta_3$ and $\beta_4$ were the regression coefficients.

**3.3.2. Spatial weight matrix.** In this paper, a geographic distance matrix and an economic-geographic nested matrix are constructed. The calculation formula is as follows:

The geographic distance matrix $W_1$: The weighted element is the reciprocal of the distance between the provincial capitals;

The economic-geographic nested matrix $W_2$:

$$W_2 = W_1 \times diag(\bar{X}_1/\bar{X}, \bar{X}_2/\bar{X}, \cdots, \bar{X}_n/\bar{X})$$

Where $diag(\ldots)$ is a diagonal matrix, and $\bar{X}_i$ is the GDP per capita of province i; $\bar{X}$ is the average of GDP per capita of all provinces. The main diagonal elements of the spatial weight matrix are 0. The geographic distance matrix and economic-geographic nested matrix can be seen in S2 and S3 Appendices.

# 4. Results and discussion

## 4.1. Spatiotemporal evolution characteristics

To reveal the evolution trend of the agro-industrial agglomeration level in provinces, we used the location entropy method to calculate the agro-industrial agglomeration degree index of 30 provinces from 2013 to 2019 and divided the 30 provinces into eastern, central, and western regions according to the regional classification criteria in the Statistical Yearbook of the National Bureau of Statistics.

There are clear regional differences in the distribution of the agro-industrial agglomeration level in China, with the overall performance being higher in the west and lower in the east. From 2013 to 2019, the average ratio of the degree of agro-industrial agglomeration in the east, middle, and west of the 30 provinces and municipalities was 1:2.35:2.53. In 2019, the ratio of the degree of agglomeration in the eastern, central, and western regions was 1:1.7:1.83. There is a large gap in the degree of agglomeration in the eastern, central, and western regions. When using location entropy to measure the degree of agro-industrial agglomeration, it takes the whole country as a whole and measures each province's relative agglomeration. Therefore, if there is a region with a relatively higher degree of agglomeration, there must be a region with a relatively lower degree of agglomeration. Therefore, from the perspective of development trends 2013–2019, the degree of agro-industrial agglomeration in 30 provinces and municipalities in China increased the fastest in Guangxi, Heilongjiang, and Inner Mongolia, and decreased the most in Beijing, Shanghai, and Anhui. Provinces and municipalities with a decreasing degree of agglomeration may have transformed their economic structure and focused on the development of secondary and tertiary industries; thus, the degree of agro-industrial agglomeration decreased. Overall, the general degree of agglomeration in the western provinces and municipalities increased, whereas that in the central and eastern regions showed a downward trend. Details are presented in Table 2.

The aggregation level of the agricultural industry was divided into three levels by quantile to further analyze its spatial distribution. In 2013, the degree of aggregation in 15 provinces and municipalities was at a low level, with nine in the eastern region, one in the central region, and five in the western region, and only Qinghai had a relatively high degree of industrial agglomeration. In 2013, the degree of agro-industrial agglomeration in 2019 was low in 13 provinces and municipalities, with eight in the east, two in the middle, and three in the west. Guangxi, Hainan, and Heilongjiang have a high degree of agricultural industry agglomeration. The degree of agglomeration in China showed an overall upward trend. The number of provinces with medium-high agglomeration levels increased, while the number of provinces with low levels decreased. The specific distributions are presented in Table 3.

**Table 2. Agglomeration of agricultural industries in the east, central and west.**

| Region | Mean | AAGR | Region | Mean | AAGR | Region | Mean | AAGR |
|---|---|---|---|---|---|---|---|---|
| Beijing | 0.07 | -11.07% | Jilin | 1.262 | 2.99% | Guangxi | 1.946 | 9.89% |
| Tianjin | 0.147 | 4.01% | Heilongjiang | 2.284 | 9.12% | Neimeng | 1.207 | 6.26% |
| Hebei | 1.342 | 0.40% | Shanxi | 0.679 | -0.05% | Chongqing | 0.9 | 0.64% |
| Shanghai | 0.054 | -8.74% | Jiangxi | 1.244 | -1.38% | Sichuan | 1.485 | -0.01% |
| Jiangsu | 0.653 | -1.95% | Anhui | 1.294 | -3.47% | Guizhou | 1.782 | 5.05% |
| Zhejiang | 0.508 | -1.83% | Henan | 1.321 | -2.63% | Yunnan | 1.841 | 0.44% |
| Fujian | 0.958 | -2.30% | Hubei | 1.331 | -2.93% | Shanxi | 1.059 | 0.51% |
| Liaoning | 1.062 | 4.32% | Hunan | 1.338 | -1.44% | Gansu | 1.599 | 1.46% |
| Shandong | 0.937 | 0.88% | | | | Qinghai | 1.16 | 4.58% |
| Guangdong | 0.552 | 0.68% | | | | Ningxia | 0.979 | 1.38% |
| Hainan | 2.802 | 1.21% | | | | Xinjiang | 1.954 | -0.89% |
| **West** | **0.571** | **-1.31%** | **Central** | **1.344** | **0.03%** | **East** | **1.447** | **1.94%** |

## 4.2. Spatial correlation analysis

Table 4 presents the global spatial Moran's I index of the agro-industrial agglomeration level of 30 provinces and municipalities from 2013 to 2019 under the economic-geography nested weights. Under the geographical and economic-geography nested weights, the global Moran's I index of the agro-industrial agglomeration level passed the significance test of 1%. This indicates that there is a certain spatial positive correlation. Therefore, the spatial effect should not be ignored.

To reveal the spatial agglomeration characteristics, a Moran's I scatter plot was drawn to identify the spatial correlation between the agglomeration level in a certain region and neighboring regions; if the distribution is in the first quadrant of Moran's I scatter plot, the region's degree of agglomeration is high, and the level of agglomeration in neighboring regions is also quite high (HH). If it is distributed in the second quadrant, this indicates that this region's level of agglomeration is low, whereas its level in neighboring regions is high (LH). If it is distributed in the third quadrant, it indicates that the level of agglomeration in this region is low, as is the level in neighboring regions (LL). If it is distributed in the fourth quadrant, it indicates that the level of agglomeration in this region is high, whereas its level in neighboring regions is low (HL). Due to limited space, this study only shows Moran's I scatter plots based on the economic-geography nested weights in 2019 (Fig 3) and 2013 (Fig 4). Most regions were located in the first and third quadrants. In 2019, the proportion of regions distributed in the first and third quadrant was 73%. In 2013, 70 proportion of the regions were distributed in the first and third quadrants. The coexistence of a high level and a low level of regional agglomeration indicates that there is a strong positive correlation in the level of agro-industrial agglomeration at the provincial level. High–high agglomeration and low–low agglomeration are the dominant types at the provincial level in China.

**Table 3. Distribution of agro-industrial agglomeration level in China in 2013 and 2019.**

| grade | range | 2013 | 2019 |
|---|---|---|---|
| Low | 0.038~1.121 | Shanghai, Beijing, Tianjin, Zhejiang, Guangdong, Jiangsu, Shanxi, Chongqing, Liaoning, Shandong, Ningxia, Fujian, Shanxi, Neimeng | Shanghai, Beijing, Tianjin, Zhejiang, Guangdong, Jiangsu, Shanxi, Fujian, Chongqing, Shandong, Ningxia, Shanxi, Anhui |
| Middle | 1.121~2.204 | Jiangxi, Jilin, Anhui, Hebei, Hubei, Hunan, Henan, Guizhou, Sichuan, Gansu, Yunnan, Guangxi, Xinjiang, Heilongjiang | Jiangxi, Hubei, Henan, Liaoning, Hunan, Hebei, Qinghai, Sichuan, Neimeng, Jilin, Gansu, Yunnan, Xinjiang, Guizhou |
| High | 2.204~3.287 | Qinghai | Guangxi, Hainan, Heilongjiang |

**Table 4. Spatial correlation analysis of agro-industry agglomeration level.**

| year | The economic-geography nested weight | | | The geographical weight | | |
|---|---|---|---|---|---|---|
| | Moran' I | Z | P | Moran' I | Z | P |
| 2019 | 0.094 | 3.407 | 0.001 | 0.086 | 3.450 | 0.001 |
| 2018 | 0.094 | 3.382 | 0.001 | 0.085 | 3.370 | 0.001 |
| 2017 | 0.089 | 3.242 | 0.001 | 0.080 | 3.206 | 0.001 |
| 2016 | 0.097 | 3.459 | 0.001 | 0.084 | 3.352 | 0.001 |
| 2015 | 0.094 | 3.357 | 0.001 | 0.079 | 3.203 | 0.001 |
| 2014 | 0.091 | 3.300 | 0.001 | 0.076 | 3.119 | 0.002 |
| 2013 | 0.090 | 3.277 | 0.001 | 0.074 | 3.050 | 0.002 |

## 4.3. Selection and results of the spatial panel regression model

**4.3.1. Impact of agro-industrial agglomeration on farmers' income.** To investigate the impact of agro-industrial agglomeration on farmers' income, a spatial panel regression analysis was conducted with farmers' income as the explained variable and the level of agro-industrial agglomeration as the explanatory variable. The following steps were adopted here: (1) First, a Lagrange multiplier test (LM test) was conducted based on the ordinary regression model, and then the robustness LM test was conducted. According to the test results, the SDM model was selected. (2) The Hausman test was performed to determine whether the fixed effects model or

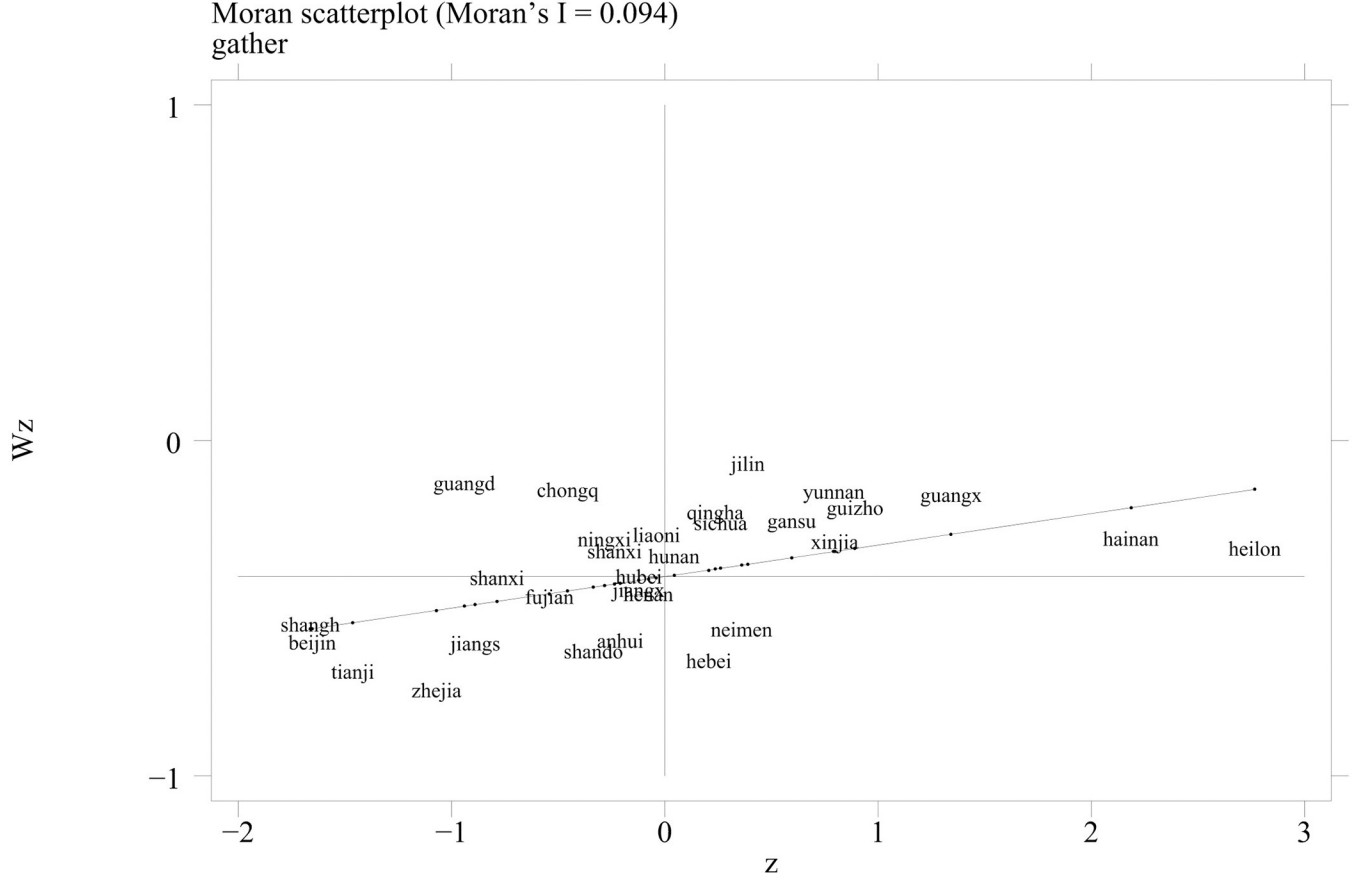

**Fig 3. Moran's I scatter plots in 2019.**

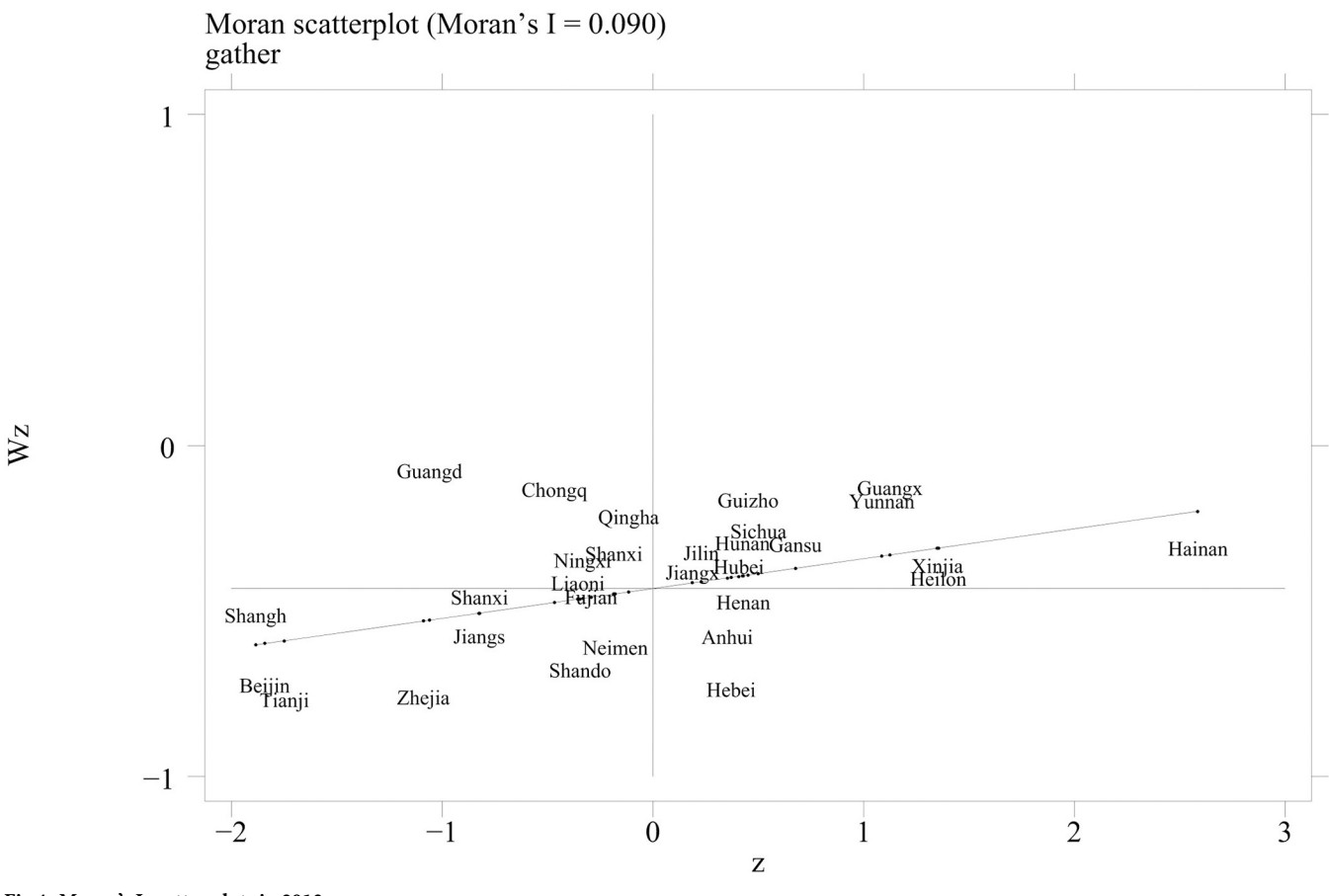

**Fig 4. Moran's I scatter plots in 2013.**

random effects model was selected. According to the results, the fixed-effect model was selected. (3) Under the fixed effects model, an LR test was conducted to determine whether the model had individual, time, or two-way fixed effects. The results show that a two-way fixed-effect model should be selected. (4) The null hypothesis is that SDM could degenerate into SEM or SAR models. The LR and WALD tests were conducted, and the null hypothesis was rejected; that is, the SDM model should be selected. Finally, after a series of tests, the SDM with two-way fixed effects was selected for the spatial panel regression analysis.

To increases the results' robustness, the ordinary panel regression model (fixed effect model) and the spatial panel SDM/SEM/SLR model were used for analysis. The regression results are presented in Table 5. The results show that, compared with the ordinary panel regression model, the SDM/SEM/SLR model considering the spatial effect has a larger maximum likelihood value and a smaller AIC value, indicating that the spatial panel model has a better effect. Among the three models in the spatial panel, the maximum likelihood value of the SDM model is the largest, and the AIC value is the smallest, which again indicates that the SDM model is superior to the other two models; however, the index difference between the SLR and the SDM is not large. The SLR model has a higher fit; therefore, the SLR model is also of great reference significance.

In the SDM, the regression coefficient of the explanatory variable agro-industrial agglomeration is 0.048, which passes the significance test of 1%. In the other three models, the agro-industrial agglomeration variable is significantly positive at the 5% level, which indicates that

**Table 5. Spatial panel model regression results.**

| | (1)Ordinary panel model | (2)SDM | (3)SEM | (4)SLR |
|---|---|---|---|---|
| | income | income | income | income |
| main | | | | |
| gather | 0.255** | 0.048*** | 0.035** | 0.035** |
| | (0.101) | (0.014) | (0.014) | (0.014) |
| GDP | 0.450*** | 0.128*** | 0.123*** | 0.121*** |
| | (0.127) | (0.023) | (0.019) | (0.020) |
| credit | 0.275*** | 0.023** | 0.031*** | 0.031*** |
| | (0.046) | (0.010) | (0.010) | (0.010) |
| edu | 0.060* | 0.008 | 0.001 | 0.001 |
| | (0.030) | (0.009) | (0.009) | (0.009) |
| invest | -0.135** | -0.019* | -0.009 | -0.009 |
| | (0.052) | (0.010) | (0.010) | (0.010) |
| machine | -0.035 | 0.029** | 0.018 | 0.018 |
| | (0.051) | (0.012) | (0.012) | (0.012) |
| _cons | 2.472** | | | |
| | (1.031) | | | |
| Wx | | | | |
| gather | | 0.212** | | |
| | | (0.099) | | |
| GDP | | 0.076 | | |
| | | (0.113) | | |
| credit | | 0.268*** | | |
| | | (0.065) | | |
| edu | | 0.072 | | |
| | | (0.047) | | |
| invest | | -0.067 | | |
| | | (0.067) | | |
| machine | | -0.099 | | |
| | | (0.073) | | |
| Spatial | | | | |
| rho | | -0.104 | | 0.107 |
| | | (0.242) | | (0.208) |
| lambda | | | 0.017 | |
| | | | (0.208) | |
| Variance | | | | |
| sigma2_e | | 0.000*** | 0.000*** | 0.000*** |
| | | (0.000) | (0.000) | (0.000) |
| N | 210.000 | 210.000 | 210.000 | 210.000 |
| r2 | 0.910 | 0.360 | 0.455 | 0.476 |
| AIC | -644.305 | -1100.555 | -1099.688 | -1100.014 |
| Log-likelihood | 328.1525 | 564.2773 | 557.8438 | 558.0071 |

**Notes**: Standard errors in parentheses.

* $p < 0.1$

** $p < 0.05$

*** $p < 0.01$.

its development contributes to an increase in farmers' income. Thus, Hypothesis 1 was verified. In terms of control variables, in the SDM model, the coefficients of economic development level, agricultural loans, and agricultural modernization are significantly positive, whereas human capital and rural fixed asset investment have no significant impact on farmers' income (the roughly same as the results of the SEM/SLR model).

According to the SDM model's results, Among the spatial interaction effects(Wx) of the explanatory variables and control variables, the two variables of agro-industrial agglomeration and agricultural loans passed the significance test, with coefficients of 0.212 and 0.268, respectively. This indicates that a 1% increase in the degree of agro-industrial agglomeration and agricultural loans level in neighboring regions will increase the income of farmers in this region by 0.212% and 0.268%, respectively. The four variables of economic development level, rural fixed asset investment, agricultural loans, and rural human capital do not have a spatial lag effect, and the empirical results are in line with the theoretical logic.

In order to assess the spillover effects, this paper decomposes the total marginal effects into direct and indirect effects. The direct effect is the spatial effect within regions, which represents the influence of explanatory variables in this region on farmers' income in this region, including the feedback effect; the indirect effect is the spatial effect between regions, which represents the influence of explanatory variables in this region on farmers' income in other regions; and the total effect is the sum of direct and indirect effects, which represents the average impact of explanatory variables in a certain region on farmers' income in all regions. Therefore, the spatial regression partial differential method is introduced to decompose the spatial effects of explanatory variables on the explained variables into direct, indirect, and total effects, as shown in Table 6.

According to the SDM and SLR model's results, the direct, indirect and total effect of agro-industrial agglomeration on farmers' income is significantly positive at the 1% level (the coefficient is 0.047, 0.194 and 0.241), indicating that there is a spacial spillover effects. This also proves that the higher degree of agro-industrial agglomeration has a significant driving effect on local and neighboring farmer income. When agro-industrial agglomeration of a region

**Table 6. Analysis of spatial effects.**

| variables | direct effects | | indirect effects | | total effects | |
|---|---|---|---|---|---|---|
| | SDM | SLR | SDM | SLR | SDM | SLR |
| gather | 0.047*** | 0.035** | 0.194** | 0.006 | 0.241*** | 0.041** |
| | (0.014) | (0.014) | (0.085) | (0.011) | (0.085) | (0.018) |
| GDP | 0.126*** | 0.120*** | 0.052 | 0.023 | 0.178** | 0.142*** |
| | (0.020) | (0.018) | (0.085) | (0.038) | (0.083) | (0.040) |
| credit | 0.022** | 0.032*** | 0.252*** | 0.005 | 0.274*** | 0.037** |
| | (0.010) | (0.010) | (0.064) | (0.011) | (0.064) | (0.015) |
| edu | 0.008 | 0.002 | 0.071 | 0.000 | 0.079 | 0.002 |
| | (0.010) | (0.010) | (0.054) | (0.003) | (0.059) | (0.012) |
| invest | -0.021* | -0.011 | -0.061 | -0.003 | -0.081 | -0.013 |
| | (0.012) | (0.011) | (0.066) | (0.007) | (0.071) | (0.016) |
| machine | 0.030*** | 0.019* | -0.090 | 0.004 | -0.060 | 0.023 |
| | (0.011) | (0.011) | (0.067) | (0.007) | (0.066) | (0.014) |

**Notes**: Standard errors in parentheses.

* $p < 0.1$

** $p < 0.05$

*** $p < 0.01$.

increases by 1%, the income of farmers in this region increases by 0.047%, the income of farmers in neighboring regions increases by 0.194%, and the income of farmers in all regions increases by 0.241% on average. (the same conclusion is reached in the SLR model.)

Regarding the control variables, agricultural loans' direct effect on farmers' income is significantly positive at the 5% level, the indirect and total effect is significantly positive at the 1% level, indicating that the improvement in the level of local agricultural loans has a driving effect on the income of local and neighboring farmers. The direct effect of economic development level and agricultural modernization on farmers' income is significantly positive at the 1% level, while the indirect and overall effects are not significant, indicating that economic development level and agricultural modernization has a driving effect on the income of local farmers but has no effect on the income of farmers in neighboring areas, which is in line with theoretical logic. The direct, indirect and overall effects of rural residents' fixed asset investment and rural human capital on farmers' income were not significant. This may be because China has more than 90% small farmers and the amount of small farmers' fixed asset investment is very small in reality, so the impact on income is not significant. In addition, China carries out nine-year compulsory education, and there is little difference in education level among provinces, so it has no significant impact on farmers' income.

**4.3.2. The Impact of agro-industrial agglomeration on farmers' income structure.** In the empirical analysis, the explained variable is farmers' income, which is measured by rural residents' disposable income. According to the classification of the China Rural Statistical Yearbook, rural residents' disposable income can be further divided into wage income, agricultural operation income, property income, and transfer income. This study, therefore, compares the proportion of wage, agricultural operation, property and transfer income in the disposable income of rural residents as explained variables, and studies the relationship between them and the independent variable agro-industrial agglomeration. The model selection is as follows: (1)Agglomeration has a spatial effect on the proportion of farmers' wage, property and transfer income, but it is not significant. (the LM coefficient is significant, but the spatial regression coefficient, the direct and indirect effect of agglomeration are not significant.) (2) Agglomeration has no spatial effect on the proportion of farmers' agricultural operation income(the LM coefficient is not significant). Therefore, ordinary panel model are used for the study.

The selection steps of the ordinary panel model are as follows: (1) the Hausman test is used to determine whether the fixed effect or random effect models should be selected; (2) If the result of Hausman test is negative, an over-identification test was performed; (3) if the fixed effects model is selected, the LSDV method is used to investigate the existence of individual fixed effects and determine whether the mixed regression or the individual fixed effects model should be selected. Meanwhile, the time fixed effect was investigated. (4) If the random effects model is selected, the LM test is used to investigate whether there are individual random effects and determine whether the mixed regression model or the individual random effects model should be selected.

After a series of tests, the fixed effect model should be selected for the model with the share of wage, operation, transfer income as the explained variable; the random effect model should be selected for the model with the share of property income as the explained variable. The empirical results are shown in Table 7.

According to these results, along with the increase in the agro-industrial concentration, farmers' income structure of the proportion of wage income will increase significantly, and a significant reduction in the agricultural operation income proportion. This also confirms that agglomeration's impact on farmers' income is mainly realized through agricultural management organizations in the agglomeration area. The increasing degree of agro-industrial

**Table 7. The influence of agricultural agglomeration on farmers' income structure.**

|  | (1)the individual fixed effects mode | (2)the two-way fixed effects mode | (3)The individual random effect mode | (4)the time fixed effects model |
|---|---|---|---|---|
|  | Proportion of wage income | Proportion of operation income | Proportion of property income | Proportion of transfer income |
| main |  |  |  |  |
| gather | 0.036** | -0.025* | 0.001 | -0.021*** |
|  | (0.017) | (0.013) | (0.002) | (0.005) |
| GDP | 0.024 | 0.039** | 0.006** | -0.083*** |
|  | (0.022) | (0.019) | (0.003) | (0.010) |
| credit | 0.016* | 0.028*** | 0.001 | -0.008 |
|  | (0.009) | (0.009) | (0.001) | (0.007) |
| edu | -0.032** | 0.012 | 0.001 | -0.022*** |
|  | (0.015) | (0.009) | (0.002) | (0.007) |
| invest | -0.023 | 0.013 | 0.002 | 0.019*** |
|  | (0.020) | (0.009) | (0.002) | (0.007) |
| machine | 0.020 | -0.036*** | -0.004*** | 0.002 |
|  | (0.016) | (0.011) | (0.001) | (0.003) |
| _cons | 0.211 | -0.127 | -0.043 | 1.453*** |
|  | (0.322) | (0.202) | (0.028) | (0.141) |
| N | 210.000 | 210.000 | 210.000 | 210.000 |
| r2 | 0.4650 | 0.3421 | 0.3176 | 0.3817 |

**Notes**: Standard errors in parentheses.

* $p < 0.1$

** $p < 0.05$

*** $p < 0.01$.

agglomeration promotes the establishment and vigorous development of agricultural enterprises, which brings more employment opportunities and higher wage levels and significantly increases the proportion of wage income in farmers' income structure. The regression results confirm that the impact of agro-industrial agglomeration on farmers' income is mainly traction effect. The proportion of agricultural operating income is decreasing, which also indicates that the economies of scale and externalities generated by agglomeration mainly apply to enterprises and have little impact on individual farmers.

**4.3.3. The impact of agro-industrial agglomeration on farmers' wage, operation, property and transfer income.** This study also uses the absolute values of four categories of farmers' disposable income as explanatory variables to analyze the impact of agro-industrial agglomeration on farmers' wage income, agricultural operation income, property income, and transfer income. The model selection is as follows: (1) Agglomeration has no spatial effect on farmers' operation income (the LM coefficient is not significant); therefore, ordinary panel data are used for the study. (2)Agglomeration has a spatial effect on farmers' wage, property and transfer income, but it is not significant (the LM coefficient is significant, but the spatial regression coefficient is insignificant, direct effect or indirect effect of agglomeration is not significant); therefore, ordinary panel data are used for the study. After a series of tests, the fixed effects model should be selected, as shown in Table 8.

According to the regression results, the agglomeration degree has a significant effect on wage, agricultural operation, property, and transfer income at the 1% level, indicating that the rise of the agro-industrial agglomeration degree has a driving effect on these types of income; that is, the driving, traction and policy effect of agro-industrial agglomeration on farmers' income are significant. To compare the differences in variable coefficients among the models,

**Table 8. The influence of agricultural agglomeration on wage, operating, property, transfer income.**

|  | Wage income | Operating income | Property income | Transfer income |
|---|---|---|---|---|
| gather | 0.378*** | 0.149*** | 0.348*** | 0.350*** |
|  | (0.062) | (0.042) | (0.113) | (0.101) |
| GDP | 0.528*** | 0.398*** | 0.570*** | 0.521*** |
|  | (0.086) | (0.058) | (0.157) | (0.139) |
| credit | 0.329*** | 0.186*** | 0.379*** | 0.343*** |
|  | (0.037) | (0.025) | (0.068) | (0.059) |
| edu | 0.008 | 0.007 | 0.101 | 0.210*** |
|  | (0.043) | (0.029) | (0.080) | (0.070) |
| invest | -0.212*** | -0.024* | -0.067 | -0.313*** |
|  | (0.046) | (0.031) | (0.084) | (0.073) |
| machine | 0.066 | -0.019 | -0.082 | -0.105 |
|  | (0.054) | (0.037) | (0.010) | (0.088) |
| _cons | 0.211 | 2.419*** | -4.182*** | -0.312 |
|  | (0.827) | (0.562) | (1.516) | (1.344) |
| N | 210.000 | 210.000 | 210.000 | 210.000 |
| r2 | 0.8518 | 0.8297 | 0.6973 | 0.7504 |
| F | 81.52*** | 90.24*** | 34.87*** | 23.38*** |

**Notes**: Standard errors in parentheses.

* $p < 0.1$

** $p < 0.05$

*** $p < 0.01$.

the coefficient difference test based on the Seemingly Unrelated Regression Estimation (SUR) was used. However, since STATA does not support such test for panel data, the individual effects in the model were first removed manually, and then the OLS estimation was used. Finally, the coefficient difference test based on the Seemingly Unrelated Regression Estimation (SUR) was used. The test results show that the coefficient of explanatory variables of the wage income model and the operation income model are significantly different. Therefore, it can be concluded that with the increase of agro-industrial agglomeration degree, the improvement of wage income is greater than the improvement of operating income. Specifically, when the degree of agglomeration increases by 1%, the wage income increases by 0.378%, and the agricultural operation income increases by 0.149%. This conclusion is similar to the above test results on the impact of agglomeration on farmers' income structure, indicating it mainly exerts a driving effect on farmers' income by influencing agricultural production organization in the agglomeration area. The traction effect should be the main role.

## 4.4. Test of traction effect

According to the theoretical analysis, we believe that the traction effect is the main role. that is, agro-industrial agglomeration mainly affects the establishment and development of agricultural organizations in the agglomeration area, and all kinds of agricultural organizations drive the growth of farmers' income in the area. Therefore, this study uses the number of agricultural operation entities in each province as a mediating variable to test the mediating effect in the theory. The stepwise regression test and bootstrap test were used to verify the mediating effect, so as to compensate for the missed detection problem caused by the opposite relationship between the mediating and independent variables or the dependent variable in the stepwise regression test.

**Table 9. Mediating effect test——the three-step test.**

|  | The two-way fixed effect model | the two-way fixed effect model | The two-way fixed effect model |
|---|---|---|---|
|  | income | firm | income |
| gather | 0.035** | 0.291** | 0.028* |
|  | (0.015) | (0.134) | (0.015) |
| firm |  |  | 0.025*** |
|  |  |  | (0.009) |
| GDP | 0.123*** | 0.055 | 0.121*** |
|  | (0.021) | (0.189) | (0.021) |
| credit | 0.031*** | 0.008 | 0.031*** |
|  | (0.011) | (0.093) | (0.010) |
| edu | 0.001 | -0.312*** | 0.009 |
|  | (0.010) | (0.087) | (0.010) |
| invest | 0.009 | -0.120** | -0.006 |
|  | (0.011) | (0.095) | (0.011) |
| machine | 0.018 | 0.207* | -0.013 |
|  | (0.013) | (0.113) | (0.013) |
| _cons | 7.450*** | 10.804*** | 7.175*** |
|  | (0.231) | (2.023) | (0.244) |
| N | 210.000 | 210.000 | 210.000 |
| r2 | 0.9899 | 0.7805 | 0.9904 |
| F | 292.60*** | 45.22*** | 300.91*** |

**Notes**: Standard errors in parentheses.

* $p < 0.1$

** $p < 0.05$

*** $p < 0.01$.

Ordinary panel model is used to test the mediating effect. The model selection method of the three-step test is the same as that of 4.3.2. According to the results, the three models of the three-step test adopt the two-way fixed effect model. The regression results are shown in Table 9. According to the regression results, the impact of agro-industrial agglomeration on agricultural operators in the area is significantly positive at the 5% level, indicating that a higher degree of agglomeration can promote the establishment and development of agricultural operation organizations. In Model (3), the two variables of agricultural industrial agglomeration and agricultural operation organizations are significantly positive, indicating that the mediating effect is established and the development of agricultural operation organizations has a significant driving effect on farmers' income.

To make the test results more robust, a bootstrap test was used to again verify the mediation effect. In the test, the bootstrap sample was selected as 1000, and the regression results are shown in Table 10. The results showed that the indirect effects (mediating effect) were significant at the 1% level, thus verifying Hypothesis 2.

Finally, this study used the same process to test the mediating effect of agricultural organizations on farmers' wage income, and agricultural operation income. The results show that agricultural organizations still have a significant mediating effect, and the test results are shown in Table 11. To compare the mediating effect strength between models (3) and (6), the individual effect in the model was manually removed, and OLS estimation was used. Finally, the coefficient difference test was based on SUR. The test results show that there are significant differences in the coefficients of agricultural operation organization variables between the

**Table 10. Mediating effect test——bootstrap.**

| | Observed | Bootstrap | | | Normal-based | |
|---|---|---|---|---|---|---|
| | Coef. | Std. Err. | z | P>\|z\| | 95% Conf. | Interval |
| _bs_1 | -0.0060 | 0.0062 | -0.98 | 0.328 | -0.018 | 0.006 |
| _bs_2 | 0.0952 | 0.0202 | 4.72 | 0.000 | 0.056 | 0.135 |

wage income model and the agricultural operating income model; that is, the mediating effect of agricultural operation organizations on wage income is greater than that on operating income, indicating that agricultural operation organizations drive farmers' income mainly by providing more job opportunities and increasing the wage level. (All Regression results and STATA do files in this paper can be seen in S4 and S5 Appendices).

## 4.5. Discussion

This paper proposes that the business subjects of agro-industrial agglomeration should be enterprises and farmers. General agglomeration theory is applicable to enterprises rather than farmers. Therefore, this paper focuses on analyzing the impact of agglomeration on farmers. In addition, previous analysis only focused on the overall impact of farmers' income variables in the statistical sense, and neither explored the underlying reasons nor analyzed the mechanisms. In this paper, the mechanism by which agricultural industry agglomeration affects farmers' income is analyzed in depth, and three effects are proposed: traction, promotion, and

**Table 11. Mediating effect test——the three-step test.**

| | (1) | (2) | (3) | (4) | (5) | (6) |
|---|---|---|---|---|---|---|
| | Wage income | Agricultural organization | Wage income | Operating income | Agricultural organization | Operating income |
| gather | 0.378*** | 0.801*** | 0.269*** | 0.148*** | 0.801*** | 0.098** |
| | (0.062) | (0.158) | (0.062) | (0.043) | (0.158) | (0.044) |
| firm | | | 0.136*** | | | 0.063*** |
| | | | (0.028) | | | (0.020) |
| GDP3 | 0.528*** | 0.846*** | 0.413*** | 0.386*** | 0.846*** | 0.333*** |
| | (0.086) | (0.219) | (0.084) | (0.059) | (0.219) | (0.060) |
| credit | 0.329*** | 0.380*** | 0.278*** | 0.187*** | 0.380*** | 0.163*** |
| | (0.037) | (0.094) | (0.036) | (0.025) | (0.094) | (0.026) |
| edu | 0.008 | -0.230** | 0.039 | 0.030 | -0.230** | 0.044 |
| | (0.043) | (0.111) | (0.041) | (0.030) | (0.111) | (0.029) |
| invest | -0.212*** | -0.436*** | -0.153*** | -0.027 | -0.436*** | 0.001 |
| | (0.046) | (0.117) | (0.045) | (0.031) | (0.117) | (0.032) |
| machine | 0.066 | -0.126 | 0.083 | -0.010 | -0.126 | -0.002 |
| | (0.054) | (0.139) | (0.051) | (0.037) | (0.139) | (0.036) |
| _cons | 0.211 | 2.686 | -0.153 | 2.319*** | 2.686 | 2.150*** |
| | (0.827) | (2.117) | (0.781) | (0.568) | (2.117) | (0.556) |
| N | 210.000 | 210.000 | 210.000 | 210.000 | 210.000 | 210.000 |
| r2 | 0.852 | 0.605 | 0.870 | 0.828 | 0.605 | 0.838 |
| F | 81.52*** | 31.86*** | 82.28*** | 89.58*** | 31.86*** | 91.13*** |

**Notes**: Standard errors in parentheses.

\* $p < 0.1$

\*\* $p < 0.05$

\*\*\* $p < 0.01$.

policy. On this basis, it is proposed that traction should be the main role; that is, agglomeration promotes the growth of farmers' income by promoting the development of enterprises in the agglomeration area. The conclusion is significant for SDGs such as ending poverty (SDG1), ending hunger (SDG2), promoting sustainable economic growth and full employment (SDG8), and building resilient infrastructure and fostering innovation (SDG9).

In the SDM, the agro-industrial agglomeration variable is significantly positive at the 1% level, indicating that its development will help increase farmers' income and achieve the SDG of no poverty (SDG1). In the group test, at the 1% level, the degree of agglomeration has a significant impact on wages, operations, property, and transfer income, indicating that agglomeration has a significant promotion, traction, and policy effect on farmers' income. The promotion effect encourages farmers to strengthen communication and learn advanced production technology because of their geographical proximity, thereby improving productivity (SDG8), increasing the output of agricultural products, and helping to achieve the sustainable goal of zero hunger (SDG2). The policy effect means that agglomeration attracts the attention of the government. The government promotes rural infrastructure construction (SDG9) through transfer payments to create a better production environment, which promotes production and increases income. The tractive effect means that agglomeration can promote the rapid development of enterprises in the agglomeration area, thereby improving the income of farmers. At the same time, the indirect effect coefficient of agglomeration is 0.194, which is significant at the 5% level, indicating that there is a spatial spillover effect on the income of farmers at the provincial level; that is, the income of farmers in this region is affected by agglomeration in the neighboring areas. Therefore, when developing local agglomeration, it can not only increase the income of farmers in this region, but also increase the income of farmers in neighboring areas.

Based on the SUR test, this study compares the effects of agglomeration on different types of income. The results show that there is a significant difference in the coefficient between salary (0.378) and operating (0.149) income. Agglomeration increases wage income more than operating income, indicating that the traction effect is the most important channel. Then, stepwise regression verifies the mediating role of agricultural organizations. The results show that the two variables of agro-industrial agglomeration (0.028) and agricultural operation entities (0.025) are significantly positive, and there is an intermediary effect. A higher degree of agglomeration can promote the establishment and development of agricultural operation entities, and agricultural operation entities play a significant role in promoting farmers' income. Therefore, the government should attach importance to the interest connection mechanism between entities and farmers, formulate policies to optimize the partnership between entities and farmers (SDG17), promote enterprises to play a leading role, and lead farmers to share the benefits of the value-added mechanism of agglomeration. For example, when formulating policies to promote agricultural operators, government departments should consider the number of farmers affected and the degree of interest linkage as important assessment indicators. Agricultural operating entities that provide technical training and guidance to farmers and help them reduce production costs and increase income are supported in terms of capital and land. The government can also encourage agricultural operators to play a leading role in information services, public goods investment, agricultural industrialization development, and other aspects through tax incentives and special subsidies, thereby improving agricultural operators' willingness to support farmers, and creating mutually beneficial, win-win interest linkage mechanisms.

Furthermore, the paper compares the mediating effects of agricultural operating entities on farmers' wage and operating income. The results show that there are significant differences in the coefficients of agricultural operation variables between the wage income model and the

agricultural operating income model at the 1% level; that is, agricultural operation entities drive farmers' income primarily by providing more job opportunities and increasing the wage level (SDG8); however, the promotion of farmers' agricultural production remains insufficient.

Agro-industrial agglomeration may also bring some potentially negative effects. For example, large-scale concentrated production may lead to environmental deterioration and reduce the resilience of farmers to natural disasters, undermining the achievement of SDG13 (climate action). In addition, due to the homogenization of agricultural products, large-scale agglomeration production may also lead to limited market competition and aggravate conflicts among communities in the agglomeration area, which is not conducive to the realization of SDG11 (sustainable cities and communities). In this regard, the government should cultivate new models, such as green agriculture, three-dimensional agriculture, circular agriculture, a combination of planting and breeding, and so on. Refined farming methods can not only improve land utilization and increase farmers' income, but also solve the environmental deterioration caused by single planting. At the same time, the government should support the establishment of agricultural cooperatives, joint production of farmers, unified quality and unified pricing; increase market competitiveness, and avoid community conflicts.

## 5. Conclusions

In this paper, the mechanism by which agricultural industry agglomeration affects farmers' income is analyzed theoretically, and three effects are proposed: traction, promotion, and policy. Then the paper uses the spatial panel model to verify the influence mechanism of agro-industrial agglomeration on farmers' income, and verifies the intermediary effect of agricultural organizations, and proves that traction is the main role. According to the results, agro-industrial agglomeration has little impact on farmers' agricultural production but primarily promotes the development of agricultural organizations, thus improving farmers' income. In the promotion of farmers' income, agricultural organizations mainly increase farmers' wage income, but has a weak effect on the promotion of farmers' agricultural operating income. Finally, the paper discusses the positive and negative effects of agro-industrial agglomeration on the Global Sustainable Development Goals, and proposes some useful suggestions.

## Supporting information

**S1 Appendix. Dataset.**
(XLS)

**S2 Appendix. The geographic distance matrix.**
(XLS)

**S3 Appendix. The economic-geographic nested matrix.**
(XLS)

**S4 Appendix. Regression results.**
(DOCX)

**S5 Appendix. STATA do file.**
(RAR)

## Author Contributions

**Conceptualization:** Yi Ding.

**Data curation:** Yi Ding.

**Investigation:** Yi Ding.

**Methodology:** Yi Ding.

**Project administration:** Yi Ding.

**Resources:** Yi Ding.

**Software:** Yi Ding.

**Writing – original draft:** Yi Ding.

**Writing – review & editing:** Yi Ding.

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
