## [Editor Report · Decision Letter 0]

3 Feb 2023

PONE-D-22-34753The impact of agricultural industrial agglomeration on farmers' income: an influence mechanism test based on a spatial panel modelPLOS ONE

Dear Dr. Ding,

Thank you for submitting your manuscript to PLOS ONE. After careful consideration, we feel that it has merit but does not fully meet PLOS ONE’s publication criteria as it currently stands. Therefore, we invite you to submit a revised version of the manuscript that addresses the points raised during the review process.

We look forward to receiving your revised manuscript.

Kind regards,

Bing Xue, Ph.D.

Academic Editor

PLOS ONE

Journal Requirements:

4. Please upload a new copy of Figure 3 as the detail is not clear. Please follow the link for more information: 

https://blogs.plos.org/plos/2019/06/looking-good-tips-for-creating-your-plos-figures-graphics/

https://blogs.plos.org/plos/2019/06/looking-good-tips-for-creating-your-plos-figures-graphics/

5. Please ensure that you refer to Figure 3 and 4 in your text as, if accepted, production will need this reference to link the reader to the figure.

6. We note you have included a table to which you do not refer in the text of your manuscript. Please ensure that you refer to Table 1 in your text; if accepted, production will need this reference to link the reader to the Table.

**Additional Editor Comments:**

An interesting study, but I dont think its ready for sending out for external review, due to the following considerations:

1) please re-format your manuscript by following the PLOS author guidlines. the current version is not friend to reviewers/

2) please draw out your conclusion based on your studies' result

3) please highlight the scientific question and your contributions.

---

## [Author Response · Author response to Decision Letter 0]

2 Apr 2023

Dear editor,

 We appreciated very much the constructive comments. In this revision, we have addressed all of these comments.

Editor Comments:

 Comment 1: please re-format your manuscript by following the PLOS author guidlines. 

 Response: Thanks for your advice. We have re-formatted the manuscript by following the PLOS author guidlines.

 Comment 2: please draw out your conclusion based on your studies' result.

 Response: Thanks for your suggestion. We have rewritten the conclusion. All conclusions are based on our studies’ result.

 Comment 3: please highlight the scientific question and your contributions.

 Response: thanks. We have highlighted the scientific question and contributions in Bold type.

---

## [Decision Letter · Decision Letter 1]

10 May 2023

PONE-D-22-34753R1The impact of agricultural industrial agglomeration on farmers' income: an influence mechanism test based on a spatial panel modelPLOS ONE

Dear Dr. Ding,

Thank you for submitting your manuscript to PLOS ONE. After careful consideration, we feel that it has merit but does not fully meet PLOS ONE’s publication criteria as it currently stands. Therefore, we invite you to submit a revised version of the manuscript that addresses the points raised during the review process.

We look forward to receiving your revised manuscript.

Kind regards,

Bing Xue, Ph.D.

Academic Editor

PLOS ONE

Journal Requirements:

Reviewers' comments:

Reviewer's Responses to Questions

**Comments to the Author**

1. If the authors have adequately addressed your comments raised in a previous round of review and you feel that this manuscript is now acceptable for publication, you may indicate that here to bypass the “Comments to the Author” section, enter your conflict of interest statement in the “Confidential to Editor” section, and submit your "Accept" recommendation.

Reviewer #1: (No Response)

2. Is the manuscript technically sound, and do the data support the conclusions?

Reviewer #1: Yes

3. Has the statistical analysis been performed appropriately and rigorously? 

Reviewer #1: Yes

4. Have the authors made all data underlying the findings in their manuscript fully available?

Reviewer #1: Yes

5. Is the manuscript presented in an intelligible fashion and written in standard English?

Reviewer #1: Yes

6. Review Comments to the Author

Reviewer #1: Clarify how the research connects with the SDG goals: The paper should clearly explain how the research relates to the SDG goals, specifically SDG 1 (no poverty) and SDG 2 (zero hunger). The authors should make a connection between the study's findings and how it contributes to these SDGs.

Highlight the role of agricultural industrial agglomeration in promoting sustainable development: The paper could highlight the role of agricultural industrial agglomeration in promoting sustainable development, specifically in the context of SDG 8 (decent work and economic growth) and SDG 9 (industry, innovation and infrastructure). The authors could emphasize how agglomeration can contribute to improving productivity and creating decent work opportunities.

Discuss the implications of the findings on policy and practice: The paper should discuss the implications of the findings on policy and practice, specifically in the context of SDG 17 (partnerships for the goals). The authors could emphasize how the findings can inform policy decisions related to promoting agricultural industrial agglomeration and improving farmers' income.

Consider the potential negative impacts of agricultural industrial agglomeration: The paper could also consider the potential negative impacts of agricultural industrial agglomeration on the environment and local communities, in the context of SDG 13 (climate action) and SDG 15 (life on land). The authors could discuss how agglomeration can lead to increased environmental degradation and social conflicts, and how these negative impacts can be mitigated.

finally send the data with STATA do file to verify the results

7. PLOS authors have the option to publish the peer review history of their article (what does this mean?). If published, this will include your full peer review and any attached files.

Reviewer #1: No

---

## [Author Response · Author response to Decision Letter 1]

21 Jun 2023

Reviewers' comments:

 comment 1: Clarify how the research connects with the SDG goals: The paper should clearly explain how the research relates to the SDG goals, specifically SDG 1 (no poverty) and SDG 2 (zero hunger). The authors should make a connection between the study's findings and how it contributes to these SDGs.

Highlight the role of agricultural industrial agglomeration in promoting sustainable development: The paper could highlight the role of agricultural industrial agglomeration in promoting sustainable development, specifically in the context of SDG 8 (decent work and economic growth) and SDG 9 (industry, innovation and infrastructure). The authors could emphasize how agglomeration can contribute to improving productivity and creating decent work opportunities.

Discuss the implications of the findings on policy and practice: The paper should discuss the implications of the findings on policy and practice, specifically in the context of SDG 17 (partnerships for the goals). The authors could emphasize how the findings can inform policy decisions related to promoting agricultural industrial agglomeration and improving farmers' income.

Response: Thanks for your advice. We have rewritten part of the introduce to clarify the importance of agro-industrial agglomeration to the Sustainable Development Goals. In addition, we have added a section "4.5discussion" to discuss the contribution of research findings to the Sustainable Development Goals (SDG1,SDG2,SDG8,SDG9,SDG17). Finally, we have proposed some useful suggestions for the policy and practice of sustainable development based on our findings.

comment 2: Consider the potential negative impacts of agricultural industrial agglomeration: The paper could also consider the potential negative impacts of agricultural industrial agglomeration on the environment and local communities, in the context of SDG 13 (climate action) and SDG 15 (life on land). The authors could discuss how agglomeration can lead to increased environmental degradation and social conflicts, and how these negative impacts can be mitigated.

Response: Thanks for your advice. In Section 4.5, we discussed the possible negative impacts of agro-industrial agglomeration, such as environmental degradation(SDG13) and community conflicts(SDG15), and proposed solutions.

comment 3: send the data with STATA do file to verify the results.

Response: Thanks for your advice. We have submitted the raw data and STATA do files in the attachment. In order to provide accurate data, we ran all the tests again. We added time and individual effect test, as well as operations such as tail reduction. Therefore, some results were slightly changed, but the overall conclusion was unchanged. Changes in the regression results are marked in red.

---

## [Editor Report · Decision Letter 2]

24 Aug 2023

The impact of agricultural industrial agglomeration on farmers' income: an influence mechanism test based on a spatial panel model

PONE-D-22-34753R2

Dear Dr. Ding,

We’re pleased to inform you that your manuscript has been judged scientifically suitable for publication and will be formally accepted for publication once it meets all outstanding technical requirements.

Kind regards,

Bing Xue, Ph.D.

Academic Editor

PLOS ONE
---

## [Editor Report · Acceptance letter]

30 Aug 2023

PONE-D-22-34753R2 

The impact of agricultural industrial agglomeration on farmers' income: an influence mechanism test based on a spatial panel model 

Dear Dr. Ding:

I'm pleased to inform you that your manuscript has been deemed suitable for publication in PLOS ONE. Congratulations! Your manuscript is now with our production department. 

Kind regards, 

on behalf of

Professor Bing Xue 

Academic Editor

PLOS ONE